# Continual Generalized Intent Discovery: Marching Towards Dynamic and Open-world Intent Recognition

**Xiaoshuai Song**[1*], **Yutao Mou**[1*], **Keqing He**[2*],**Yueyan Qiu**[1],**Pei Wang**[1], **Weiran Xu**[1*]

[1]Beijing University of Posts and Telecommunications, Beijing, China

[2]Meituan, Beijing, China

{songxiaoshuai,myt,yuanqiu,wangpei,xuweiran}@bupt.edu.cn

hekeqing@meituan.com

## Abstract

In a practical dialogue system, users may input out-of-domain (OOD) queries. The Generalized Intent Discovery (GID) task aims to discover OOD intents from OOD queries and extend them to the in-domain (IND) classifier. However, GID only considers one stage of OOD learning, and needs to utilize the data in all previous stages for joint training, which limits its wide application in reality. In this paper, we introduce a new task, Continual Generalized Intent Discovery (CGID), which aims to continuously and automatically discover OOD intents from dynamic OOD data streams and then incrementally add them to the classifier with almost no previous data, thus moving towards dynamic intent recognition in an open world. Next, we propose a method called Prototype-guided Learning with Replay and Distillation (PLRD) for CGID, which bootstraps new intent discovery through class prototypes and balances new and old intents through data replay and feature distillation. Finally, we conduct detailed experiments and analysis to verify the effectiveness of PLRD and understand the key challenges of CGID for future research.[1]

## 1 Introduction

The traditional intent classification (IC) in a task-oriented dialogue system (TOD) is based on a closed set assumption (Chen et al., 2019; Yang et al., 2021; Zeng et al., 2022) and can only handle queries within a limited scope of in-domain (IND) intents. However, users may input out-of-domain (OOD) queries in the real open world. Recently, the research community has paid more attention to OOD problems. OOD detection (Lin and Xu, 2019; Zeng et al., 2021; Wu et al., 2022; Mou et al., 2022d) aims to identify whether a user's query is outside the range of the predefined intent set to

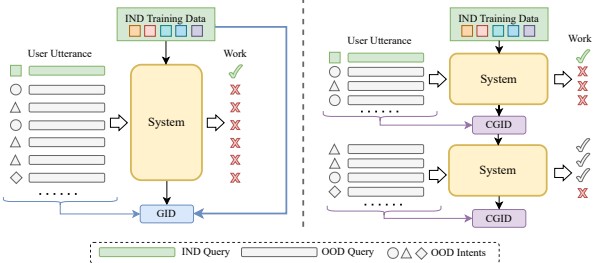

Figure 1: Illustration of the defects of GID and the advantages of CGID. GID only perform single stage of OOD learning and requires all IND data for joint training. In contrast, CGID timely updates the system from dynamic OOD data streams through continual OOD learning stage and almost does not rely on previous data.

avoid wrong operations. It can safely reject OOD intents, but it also ignores OOD concepts that are valuable for future development. OOD intent discovery (Lin et al., 2020; Zhang et al., 2021; Mou et al., 2022c,a) helps determine potential development directions by grouping unlabelled OOD data into different clusters, but still cannot incrementally expand the recognition scope of existing IND classifiers. Generalized Intent Discovery (GID) (Mou et al., 2022b) further trains a network that can classify a set of labelled IND intent classes and simultaneously discover new classes from an unlabelled OOD set and incrementally add them to the classifier.

Although GID realizes the incremental expansion of the recognition scope of the intent classifier without any new intents labels, two major problems limit the widespread application of GID in reality as shown in Fig 1: (1) **GID only considers single-stage of OOD discovery and classifier expansion.** In real scenarios, OOD data is gradually collected over time. Even if the current intent classifier is incrementally expanded, new OOD queries and intents will continue to emerge. Besides, the timeliness of OOD discovery needs to be considered:

---

*The first three authors contribute equally. Weiran Xu is the corresponding author.

[1]We release our code at https://github.com/songxiaoshuai/CGID

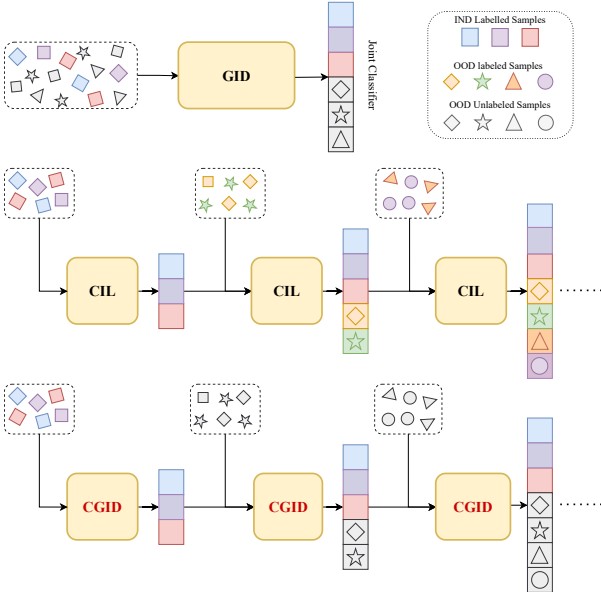

Figure 2: The comparison of CGID with GID and CIL.

timely discovery of new intents and expansion to the system can help improve the subsequent user experience. (2) **GID require data in all previous stages for joint training to maintain the classification ability for known intents.** Since OOD samples are collected from users' real queries, storing past data may bring serious privacy issues. In addition, unlike Class Incremental Learning (CIL) that require new classes with real labels, it is hard to obtain a large amount of dynamic labeled data in reality, and the label set for OOD queries is not predefined and needs to be mined from query logs.

Inspired by the above problems, in this paper, we introduce a new task, **C**ontinual **G**eneralized **I**ntent **D**iscovery (**CGID**), which aims to continually and automatically discover OOD intents from OOD data streams and expand them to the existing IND classifier. In addition, CGID requires the system to maintain the ability to classify known intents with almost no need to store previous data, which makes existing GID methods fails to be applied to CGID. Through CGID, the IC system can continually enhance the ability of intent recognition from unlabeled OOD data streams, thus moving towards dynamic intent recognition in an open world. We show the difference between CGID and GID, as well as the CIL task in Fig 2 and then leave the definition and evaluation protocol in Section 2.

As CGID needs to continuously learn from unlabeled OOD data, it is foreseeable that the system will inevitably suffer from the catastrophic forgetting (Biesialska et al., 2020; Masana et al., 2022) of

known knowledge as well as the interference and propagation of OOD noise (Wu et al., 2021). To address the issues, we propose the **P**rototype-guided **L**earning with **R**eplay and **D**istillation (PLRD) for CGID. Specifically, PLRD consists of a main module composed of an encoder and a joint classifier, as well as three sub-modules: (1) class prototype guides pseudo-labels for new OOD samples and alleviate the OOD noise; (2) feature distillation reduces catastrophic forgetting; (3) a memory balances new class learning and old class classification by replaying old class samples (Section 3) [2]. Furthermore, to verify the effectiveness of PLRD, we construct two public datasets and three baseline methods for CGID. Extensive experiments prove that PLRD has significant performance improvement and the least forgetting compared to the baselines, and achieves a good balance among old classes classification, new class discovery and incremental learning (Section 4). To further shed light on the unique challenges faced by the CGID task, we conduct detailed qualitative analysis (Section 5). We find that the main challenges of CGID are conflicts between different sub-tasks, OOD noise propagation, fine-grained OOD classes and strategies for replayed samples (Section 6), which provide profound guidance for future work.

Our contributions are three-fold: (1) We introduce a new task, Continual Generalized Intent Discovery (CGID), to achieve the dynamic and open-world intent recognition and then construct datasets and baselines for evaluating CGID. (2) We propose a practical method PLRD for CGID, which guides new samples through class prototypes and balances new and old tasks through data replay and feature distillation. (3) We conduct comprehensive experiments and in-depth analysis to verify the effectiveness of PLRD and understand the key challenges of CGID for future work.

## 2 Problem Definition

In this section, we first briefly introduce the Generalized Intent Discovery(GID) task, then delve into the details of the Continual Generalized Intent Discovery (CGID) task we proposed.

---

[2] PLRD's memory mechanism stores only a tiny fraction of samples, offering a significant privacy advantage compared to GID, which stores all past data. PLRD serves not only to provide a privacy-conscious mode of learning but also takes into account the long-term stability of task performance. As for the methods that completely eliminate the need for prior data storage, we leave them for further exploration.

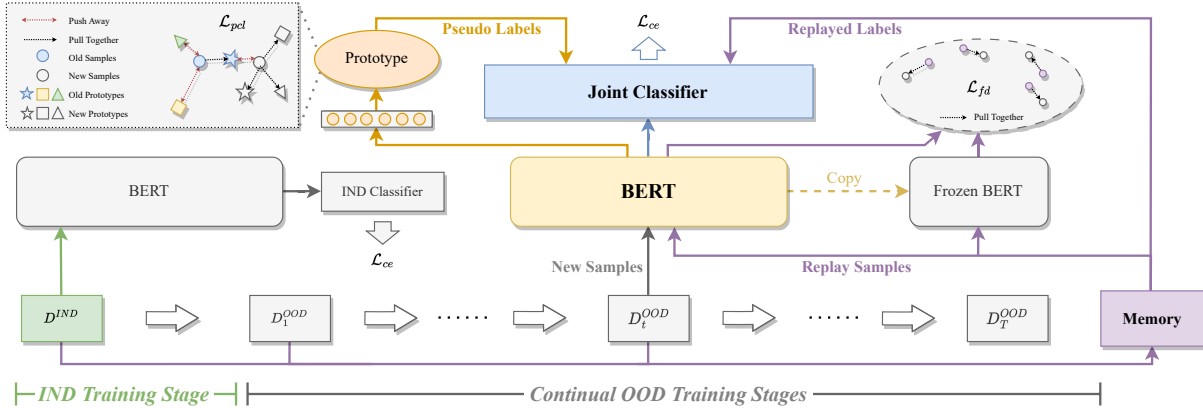

Figure 3: The Overall architecture of our PLRD method. During the IND training stage, we only use the cross-entropy loss. In the OOD training stages, multiple modules and learning objectives jointly optimize the model.

## 2.1 GID

Given a set of labeled in-domain data $D^{IND} = \{(x_i^{IND}, y_i^{IND})\}_{i=1}^n$ and unlabeled OOD data $D^{OOD} = \{(x_i^{OOD})\}_{i=1}^m$, where $y_i^{IND} \in Y^{IND}, Y^{IND} = \{1, 2, \ldots, N\}$, GID aims to train a joint classifier to classify an input query into the total label set $Y = \{1, \ldots, N, N+1, \ldots \ldots, N+M\}$, where the first N elements represent the labeled IND classes and the last M elements represent newly discovered unlabeled OOD classes.

## 2.2 CGID

In contrast, CGID provides data and expands the classifier in a sequential manner, which is more in line with real scenarios.

First, we define $t \in [0, T]$, which denotes the current learning stage of CGID and $T$ denotes the maximum number of learning stages of CGID. In the IND learning stage ($t = 0$), given a labeled in-domain dataset $D^{IND} = \{(x_i^{IND}, y_i^{IND})\}$, the model needs to classify IND classes to a predefined set $Y_0 = \{1, 2, \ldots, N\}$ and learn representations that are also helpful for subsequent stages.

Then, a series of unlabeled out-of-domain datasets $\{D_t^{OOD}\}_{t=1}^T$ are given in sequence, where $D_t^{OOD} = \{x_i^{OOD_t}\}$. In the OOD learning stage $t \in [1, T]$, the model is expected to discover new OOD classes $Y_t$[3] from $D_t^{OOD}$ and incrementally extend $Y_t$ to the classifier while maintaining the ability to classify known classes $\{Y_{i \leq t}\}$. The ultimate goal of CGID is to obtain a joint classifier that can classify queries into the total label set $Y_T^{all} = Y_0 \cup Y_1 \cup \ldots \cup Y_T$.

## 2.3 Evaluation Protocol

For CGID, we mainly focus on the classification performance along the training phase. Following (Mehta et al., 2021), we let $a_{t,i}$[4] denote the accuracy on class set $Y_i$ after training on stage $t$. When $t > 0$, we calculate the accuracy $A_t$ as follows:

$$A_t^{IND} = a_{t,0} \quad A_t^{OOD} = \frac{1}{|\{Y_{1 \leq i \leq t}\}|} \sum_{i=1}^t |Y_i| a_{t,i}$$

$$A_t^{ALL} = \frac{1}{|Y_t^{all}|} \sum_{i=0}^t |Y_i| a_{t,i}$$

(1)

Moreover, to measure catastrophic forgetting in CGID, we introduce the forgetting $F_t$ as follows:

$$F_t^{IND} = a_{0,0} - a_{t,0}$$

$$F_t^{OOD} = \frac{1}{|\{Y_{1 \leq i \leq t}\}|} \sum_{i=1}^t |Y_i|(a_{i,i} - a_{t,i})$$

$$F_t^{ALL} = \frac{1}{|Y_t^{all}|} \sum_{i=0}^t |Y_i|(a_{i,i} - a_{t,i})$$

(2)

On the whole, $A_t^{IND}$ and $F_t^{IND}$ measure the extent of maintaining the IND knowledge, while $A_t^{OOD}$ and $F_t^{OOD}$ denote the ability to learn the OOD classes. $A_t^{ALL}$ and $F_t^{ALL}$ are comprehensive metrics for CGID.

## 3 Method

### 3.1 Overall Architecture

As shown in Fig 3, our proposed PLRD framework consists of a main module which composed

---

[3]Estimating $|Y_t|$ is out of the scope of this paper. In the following experiment, we assume that $|Y_t|$ is ground-truth and provide an analysis in Section 5.5.

[4]Following (Zhang et al., 2021), we use the Hungarian algorithm (Kuhn, 1955) to obtain the mapping between the predicted OOD classes and ground-truth classes in the test set.

of an encoder and a joint classifier and three sub-modules: (1) **Memory module** is responsible for replaying known class samples to balance the learning of new classes and maintain known classes; (2) **Class prototype module** is responsible for generating pseudo-labels for new OOD samples; (3) **Feature distillation** is responsible for alleviating catastrophic forgetting of old classes. The joint classifier $h$ consists of an old class classification head $h^{old}$ and a new class classification head $h^{new}$, outputting logit $l = [l^{old}; l^{new}]$. After stage $t$ ends, $l^{new}$ will be merged into $l^{old}$, i.e., $l^{old} \leftarrow [l^{old}; l^{new}]$. Then, when stage $t + 1$ starts, a new head $l^{new}$ with the dimension $|Y_{t+1}|$ will be created.

## 3.2 Memory for Data Replay

We equip a memory module $M$ for PLRD. After each learning stage, $M$ stores a very small number of training samples and replays old class samples in the next learning stage to prevent catastrophic forgetting and encourage positive transfer. Specifically, in the IND learning stage, we randomly select $n$ samples for each IND class according to the ground-truth labels; in each OOD learning stage, since the ground-truth labels are unknown, we randomly select $n$ samples[5] for each new class according to the pseudo-labels and store them in $M$ together with the pseudo-labels. In the new learning stage, for each batch, we randomly select old class samples $\{x^{old}\}$ with the same number as new class samples $\{x^{new}\}$ from $M$ and input them into the BERT encoder $f(\cdot)$ together with new class samples $x^{new}$, i.e., $|\{x^{new}\}| = |\{x^{old}\}|$, $\{x\} = \{x^{new}\} \cup \{x^{old}\}$.

## 3.3 Prototype-guide Learning

Previous semantic learning studies (Yu et al., 2020; Wang et al., 2022; Ma et al., 2022; Dong et al., 2023a,b) have shown that learned representations can help to disambiguate the noisy sample labels and mitigate forgetting. Therefore, we build prototypes through a linear projection layer $g(\cdot)$ after the encoder. In stage $t > 0$, we first randomly initialize new class prototype $\mu_j, j \in Y_t$.[6] For sample $x_i \in \{x\}$, we use an $|Y_t^{all}|$-dimensional vector $q_i$ representing the probabilities of $x_i$ being assigned

---

[5]In the following experiment, we set $n$=5 and analyze the effects of different $n$ in the Section 5.3.

[6]When $t = 1$, we additionally initialize the prototypes of IND classes.

| Stage | Banking OOD Ratio | | | CLINC OOD Ratio | | |
|---|---|---|---|---|---|---|
| | 40% | 60% | 80% | 40% | 60% | 80% |
| 0 | 47 | 32 | 17 | 90 | 60 | 30 |
| 1 | 10 | 15 | 20 | 20 | 30 | 40 |
| 2 | 10 | 15 | 20 | 20 | 30 | 40 |
| 3 | 10 | 15 | 20 | 20 | 30 | 40 |

Table 1: The number of new classes at each stage.

to all prototypes:

$$q_i = \begin{cases} \text{onehot}(y_i^{old}) & x_i \in \{x^{old}\} \\ \left[\mathbf{0}_{|Y_{t-1}^{all}|}; l_i^{new}\right] & x_i \in \{x^{new}\} \end{cases} \quad (3)$$

where $y_i^{old}$ is the ground-truth or pseudo label of $x_i$ in $M$ and $\mathbf{0}_{|Y_{t-1}^{all}|}$ is a $|Y_{t-1}^{all}|$-dimensional zero vector. Then we introduce prototypical contrastive learning (PCL) (Li et al., 2020) as follows:

$$\mathcal{L}_{pcl} = -\sum_{i,j} q_i^j \log \frac{\exp(\text{sim}(z_i, \mu_j)/\tau)}{\sum_r \exp(\text{sim}(z_i, \mu_r))/\tau} \quad (4)$$

where $\tau$ denotes temperature, $q_i^j$ is the $j$-th element of $q_i$ and $z_i = g(f(x_i))$. By pulling similar samples into the same prototype, PCL can learn clear intent representations for new classes and maintain representations for old classes. To further improve the generalization of representation, we also introduce the instance-level contrastive loss (Chen et al., 2020) to $x_i$ as follows:

$$\mathcal{L}_{ins} = -\sum_i \log \frac{\exp(\text{sim}(z_i, \hat{z}_i)/\tau)}{\sum_j \mathbf{1}_{[i \neq j]} \exp(\text{sim}(z_i, z_j)/\tau)} \quad (5)$$

where $\hat{z}_i$ denotes the dropout-augmented view of $z_i$. Next, we update all new and old prototypes in a sample-wise moving average manner to reduce the computational complexity following (Wang et al., 2022). For sample $x_i$, prototype $\mu_j$ is updated as follows:

$$\mu_j = \gamma\mu_j + (1 - \gamma)z_i \quad (6)$$

where the moving average coefficient $\gamma$ is an adjustable hyperparameter and $j$ is the index of the maximum element in $q_i$.

Finally, for the new sample $x_i \in \{x^{new}\}$, its pseudo label is assigned as the index of the nearest new class prototype to its representation $z_i$. We optimize the joint classifier using cross-entropy $\mathcal{L}_{ce}$ over both the new and replayed samples.

| Method | OODRatio = 40% | | | | | | OODRatio = 60% | | | | | | OODRatio = 80% | | | | | |
|---|---|---|---|---|---|---|---|---|---|---|---|---|---|---|---|---|---|---|
| | IND | | OOD | | ALL | | IND | | OOD | | ALL | | IND | | OOD | | ALL | |
| | $A_T \uparrow$ | $F_T \downarrow$ | $A_T \uparrow$ | $F_T \downarrow$ | $A_T \uparrow$ | $F_T \downarrow$ | $A_T \uparrow$ | $F_T \downarrow$ | $A_T \uparrow$ | $F_T \downarrow$ | $A_T \uparrow$ | $F_T \downarrow$ | $A_T \uparrow$ | $F_T \downarrow$ | $A_T \uparrow$ | $F_T \downarrow$ | $A_T \uparrow$ | $F_T \downarrow$ |
| Kmeans | 77.94 | 15.64 | 72.00 | 14.08 | 75.63 | 15.03 | 72.55 | 22.13 | 62.55 | 12.67 | 66.71 | 16.60 | 64.61 | 31.42 | 49.64 | 15.10 | 52.94 | 18.71 |
| DeepAligned | 77.91 | 15.67 | 73.39 | 12.13 | 76.15 | 14.29 | 72.16 | 22.52 | 65.02 | 12.50 | 67.98 | 16.67 | 68.04 | 27.99 | 58.78 | 12.19 | 60.81 | 15.67 |
| E2E | 74.20 | 19.38 | 74.91 | 14.92 | 74.48 | 17.64 | 71.28 | 23.41 | 69.44 | 13.06 | 70.20 | 17.36 | 65.59 | 30.44 | 62.81 | 15.48 | 63.41 | 18.78 |
| PLRD (Ours) | 83.94 | 9.64 | 76.70 | 11.58 | 81.11 | 10.40 | 81.07 | 13.62 | 70.30 | 10.69 | 74.77 | 11.91 | 76.23 | 19.80 | 63.19 | 11.19 | 66.06 | 13.09 |

Table 2: Performance comparison on Banking after the final stage $T$=3. $\uparrow$ indicates higher is better, $\downarrow$ indicates lower is better. We bold the best results and underline the second-best results. Results are averaged over three random run (p < 0.01 under t-test).

| Method | OOD Ratio=40% | | | | | | OOD Ratio=60% | | | | | | OOD Ratio=80% | | | | | |
|---|---|---|---|---|---|---|---|---|---|---|---|---|---|---|---|---|---|---|
| | IND | | OOD | | ALL | | IND | | OOD | | ALL | | IND | | OOD | | ALL | |
| | $A_T \uparrow$ | $F_T \downarrow$ | $A_T \uparrow$ | $F_T \downarrow$ | $A_T \uparrow$ | $F_T \downarrow$ | $A_T \uparrow$ | $F_T \downarrow$ | $A_T \uparrow$ | $F_T \downarrow$ | $A_T \uparrow$ | $F_T \downarrow$ | $A_T \uparrow$ | $F_T \downarrow$ | $A_T \uparrow$ | $F_T \downarrow$ | $A_T \uparrow$ | $F_T \downarrow$ |
| Kmeans | 91.38 | 6.82 | 85.37 | 5.55 | 88.98 | 6.31 | 89.29 | 9.56 | 81.58 | 5.67 | 84.66 | 7.23 | 81.93 | 17.11 | 72.94 | 6.36 | 74.74 | 8.51 |
| DeepAligned | 91.98 | 6.22 | 89.45 | 4.61 | 90.96 | 5.58 | 88.04 | 10.82 | 86.15 | 5.04 | 86.91 | 7.35 | 84.30 | 14.74 | 80.59 | 6.25 | 81.33 | 7.95 |
| E2E | 89.16 | 9.04 | 90.74 | 7.33 | 89.79 | 8.36 | 84.96 | 13.89 | 88.44 | 6.82 | 87.05 | 9.65 | 79.33 | 19.70 | 85.74 | 7.95 | 84.46 | 10.30 |
| PLRD (Ours) | 93.90 | 4.29 | 91.22 | 5.00 | 92.83 | 4.58 | 92.15 | 6.70 | 89.09 | 3.78 | 90.31 | 4.95 | 89.56 | 9.48 | 84.76 | 5.33 | 85.72 | 6.16 |

Table 3: Performance comparison on CLINC.

## 3.4 Feature Distillation

It can be expected that the encoder features may change significantly when updating the network parameters in the new learning stage. This means that the network tends to forget the knowledge learned from the old classes before and suffers from catastrophic forgetting. To further remember the knowledge in the non-forgotten features, we integrate the feature distillation into PLRD. Specifically, at the beginning of stage $t$, we copy and freeze the encoder, denoted as $f^{init}(\cdot)$. Then given replayed samples $x_i \in \{x^{old}\}$ in a batch, we constrain the feature output $f(x_i)$ of the current encoder with the feature $f^{init}(x_i)$. Formally, the feature distillation loss is as follows:

$$\mathcal{L}_{fd} = \sum_{i=1}^{|\{x^{old}\}|} (f(x_i) - f^{init}(x_i))^2 \quad (7)$$

## 3.5 Overall Training

The total loss is defined as follows:

$$\mathcal{L} = \mathcal{L}_{ce} + \mathcal{L}_{pcl} + \mathcal{L}_{ins} + \mathcal{L}_{fd} \quad (8)$$

## 4 Experiment

### 4.1 Datasets

We construct the CGID datasets based on two widely used intent classification datasets, Banking (Casanueva et al., 2020) and CLINC (Larson et al., 2019). Banking covers only a single domain, containing 13,083 user queries and 77 intents, while CLINC contains 22,500 queries covering 150 intents across 10 domains. For the CLINC and Banking datasets, we randomly select a specified proportion of all intent classes (about 40%, 60%, and 80%

respectively) as OOD types, with the rest being IND types. Furthermore, we assign the maximum stage $T$=3, so we divide the OOD data into three equal parts for each OOD training stage. We show the number of classes at each stage in Table 1 and leave the detailed statistics in Appendix A.

### 4.2 Baselines

Since this is the first study on CGID, there are no existing methods that solve exactly the same task. We adopt three prevalent methods in OOD discovery and GID, and extend them to the CGID setting to develop the following competitive baselines[7].

- **K-means** is a pipeline baseline which first use the clustering algorithm K-means (MacQueen, 1965) to cluster the new samples to obtain pseudo labels and then combine these samples and replayed samples in the memory to train the joint classifier at each OOD training stage.

- **DeepAligned** is another pipeline baseline that leverages the iterative clustering algorithm DeepAligned (Zhang et al., 2021). At each OOD training phase, DeepAligned iteratively clusters the new data and then utilizes them along with the replayed samples for classification training.

- **E2E** is an end-to-end baseline. At each OOD training stage, E2E (Mou et al., 2022b) amalgamates the new instances and replayed samples and then obtain the logits through the encoder and joint classifier. The model is optimized with a unified classification loss, where the new OOD pseudo-labels are obtained by swapping predictions (Caron et al., 2020).

---

[7]We leave the detailed implementation of these baselines in Appendix B.

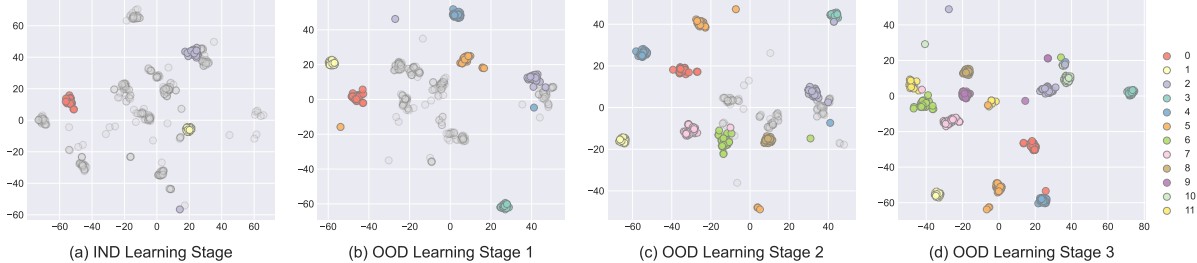

Figure 4: Intents visualization of different learning stage under Banking OOD ratio = 40% (index 0-2 belongs to $Y_0$, index 3-5 belongs to $Y_1$, index 6-8 belongs to $Y_2$ and index 9-11 belongs to $Y_3$).

## 4.3 Main Results

We conduct experiments on Banking and CLINC with three different OOD ratios, as shown in Tables 2 and 3. In general, our proposed PLRD consistently outperform all the baselines with a large margin. Next, we analyze the results from three aspects:

(1) **Comparison of different methods** We observe that DeepAligned roughly achieves the best IND performance while E2E has the best OOD performance among all baselines. However, our proposed PLRD consistently outperforms all baselines significantly in both IND and OOD, achieving best performance and new-old task balance. Specifically, under the average of three ratios, PLRD is better than the optimal baseline by 7.57% ($A_T^{IND}$), 1.09% ($A_T^{OOD}$), and 4.06% ($A_T^{ALL}$) on Banking, and by 3.35% ($A_T^{IND}$), 0.04% ($A_T^{OOD}$), and 2.13% ($A_T^{ALL}$) on CLINC. As for forgetting, E2E experiences a substantial performance drop on old classes when learning new classes, while PLRD is lower than the optimal baseline by 3.72%, 1.69% ($F_T^{ALL}$) on Banking and CLINC respectively. This indicates PLRD does not sacrifice too much performance on old classes when learning new classes and has the least forgetting among all methods.

(2) **Comparison of different datasets** We validate the effectiveness of our method on different datasets, where CLINC is multi-domain and coarse-grained while Banking contains more fine-grained intent types within a single domain. We see that the performance of all methods on CLINC is significantly better than that on Banking. For example, PLRD is 11.72% ($A_T^{ALL}$) higher on CLINC than on Banking at an OOD ratio of 60%. In addition, at the same OOD ratio, PLRD shows an average increase of 7.53% ($F_T^{IND}$), 6.45% ($F_T^{OOD}$), and 6.57% ($F_T^{ALL}$) on Banking over CLINC. We believe this could be because fine-grained new and old classes

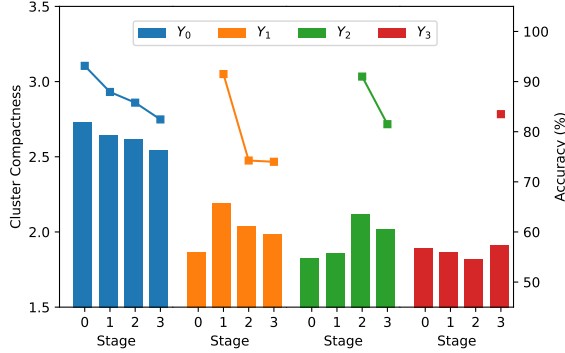

Figure 5: The compactness and accuracy of classes at different stages under Banking OOD ratio = 40%.

are easily confused, which leads to serious new-old task conflicts and high forgetting. However, PLRD achieves larger improvements than baselines on Banking, indicating that PLRD can better cope with challenges in fine-grained intent scenarios.

(3) **Effect of different OOD ratios** We observe that as the OOD ratio increases, the forgetting of IND classes increases and accuracy of OOD classes decreases significantly for all methods. For PLRD, when the OOD ratio increases from 40% to 80% on Banking, $F_T^{IND}$ rises from 9.64% to 19.80%, and $A_T^{OOD}$ drops from 76.70% to 63.19%. Intuitively, more OOD classes make it challenging to distinguish samples from different distributions, leading to noisy pseudo-labels. Moreover, in the incremental process, more OOD classes will update the model to a greater extent, resulting in more IND knowledge forgetting.

## 5 Qualitative Analysis

### 5.1 Representation Learning

In order to better understand the evolution of CGID along different training stages, we visualize the intent representations after each training stage for PLRD in Fig 4. It can be seen that in the IND learn-

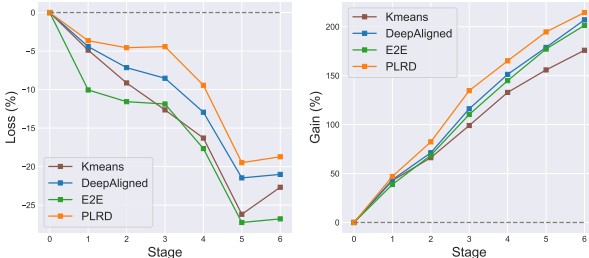

Figure 6: The Loss and Gain of the classifier at different stage under Banking, where the maximal stage $T=6$, the number of IND classes is 17, and the number of new classes in each OOD stage is 10.

ing stage, the IND classes form compact clusters, while the OOD samples are scattered in space. As the stage progresses, the gray points are gradually colored and move from dispersion to aggregation, indicating that new OOD classes continue to be discovered and learned good representations. In addition, the already aggregated clusters are gradually dispersed (see "red" points), indicating that the representations of old classes are deteriorating.

Next, to quantitatively measure the quality of representations, we calculate the intra-class and inter-class distances and use the ratio of inter-class distance to intra-class distance as the compactness following (Islam et al., 2021). We report the compactness and accuracy in Fig 5. It can see that the compactness of OOD classes is much lower than that of IND classes, indicating that the representation learning with labeled IND samples outperforms that with unlabeled OOD samples. As the stage $t$ increases, the compactness of the IND classes gradually decreases. And the compactness of the $Y_i(i > 0)$ classes increases significantly when $t$ equals $i$, and then gradually decreases. This demonstrates the learning and forgetting effects in CGID from a representation perspective. Furthermore, we observe that the maximal compactness of $Y_i$ decreases as $i$ increases, showing that the learning ability of new classes gradually declines. We attribute this to the noise in the OOD pseudo-labeled data and the greater need to suppress forgetting of more old classes. Finally, the trend of accuracy and compactness remains consistent, suggesting that representation is closely related to the classification performance of CGID.

## 5.2 Loss and Gain of CGID

During the CGID process, the performance of the classifier on IND classes gradually declined, while

| Strategy | Replay | | $A_T^{IND}$ | $A_T^{OOD}$ | $A_T^{ALL}$ |
|---|---|---|---|---|---|
| | Acc | Var | | | |
| *random* | 77.78 | 0.31 | 80.39 | **70.00** | **74.32** |
| *icarl* | **84.45** | 0.10 | **80.55** | 69.78 | 74.25 |
| *icarl_contrary* | 48.44 | **0.62** | 78.44 | 61.00 | 68.25 |

Table 4: Comparison of different selection strategies under Banking OOD ratio = 60%.

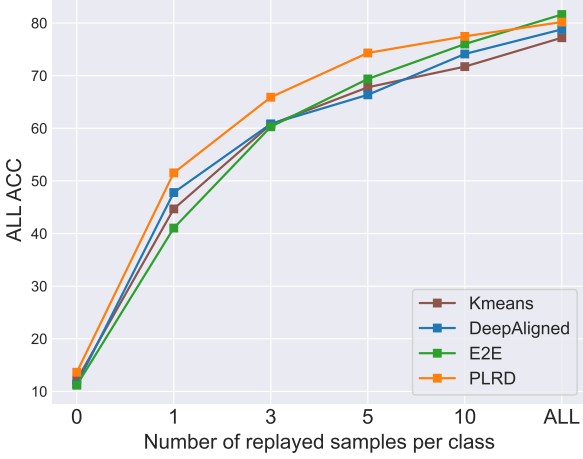

Figure 7: The effect of different number of replayed samples under Banking OOD ratio = 60%, where *ALL* means storing all training data in the memory.

the number of supported OOD classes continually expanded. In order to quantify the change of the classifier, we define the Loss and the Gain in stage $t$ for CGID as follows:

$$Loss_t = \frac{-F_t^{IND}}{A_0^{IND}} \quad Gain_t = \frac{|Y_t^{all}|A_t^{ALL}}{|Y_0|A_0^{IND}} - 1 \tag{9}$$

We illustrate the variations in Loss and Gain of all methods over stages in Fig 6. The results show that as the training progresses, the Loss of all methods decreases overall and the Gain increases continuously. After finishing the training, although the Loss decrease by 20% roughly, the increase in Gain of PLRD is greater, reaching over 200%. This indicates that the Gain generated by CGID is much higher than the Loss and brings positive effects to the classifier as a whole. Compared with other methods, PLRD has the lowest Loss and highest Gain at each stage, and its advantage continuous amplifies over stages. These further consolidate the conclusion that PLRD outperforms the baselines.

## 5.3 Effect of Replaying Samples in Memory

In this section, we explore the effect of replayed samples from both selection strategies and quantity.

**Selection Strategy** In the CIL task, since the

| Model | $A_T^{IND}$ | $A_T^{OOD}$ | $A_T^{ALL}$ |
|---|---|---|---|
| E2E | 71.28 | 69.44 | 70.20 |
| PLRD | **80.39** | **70.00** | **74.32** |
| -w/o $\mathcal{L}_{fd}$ | 75.94 | 66.78 | 70.58 |
| -w/o $\mathcal{L}_{ins}$ | 79.77 | 68.17 | 72.99 |
| -w/o $\mathcal{L}_{pcl}$ | 80.16 | 69.50 | 73.93 |
| $\mathcal{L}_{ce}$ | 73.36 | 63.67 | 67.69 |

Table 5: Ablation study of different learning objective for PLRD under Banking OOD ratio = 60%.

| | Method | Num of Classes | | | $A_T^{IND}$ | $A_T^{OOD}$ | $A_T^{ALL}$ |
|---|---|---|---|---|---|---|---|
| | | t=1 | t=2 | t=3 | | | |
| Ground Truth | DeepAligned | 15 | 15 | 15 | 72.16 | 65.02 | 67.98 |
| | E2E | 15 | 15 | 15 | 71.28 | 69.44 | 70.20 |
| | PLRD | 15 | 15 | 15 | **81.07** | **70.30** | **74.77** |
| Estimate by model itself | DeepAligned | 13 | **14** | 7 | 71.25 | 49.56 | 58.57 |
| | E2E | 13 | 12 | 11 | 67.03 | 55.33 | 60.19 |
| | PLRD | **13** | **14** | **12** | **79.22** | **66.28** | **71.66** |
| Estimate by IND Model | DeepAligned | 13 | 12 | 11 | 72.42 | 56.94 | 63.38 |
| | E2E | 13 | 12 | 11 | 67.03 | 55.33 | 60.19 |
| | PLRD | 13 | 12 | 11 | **79.22** | **60.72** | **68.41** |

Table 6: Estimate the number of new classes at each OOD learning stage under Banking OOD ratio = 60%.

samples are labeled, we only need to consider the diversity of replayed samples. However, in CGID, we need to take the quality of pseudo-labels into account additionally. We explore three selection strategies for replaying samples: *random* (randomly sampling from training set), *icarl* (selecting these closest to their prototypes, following Rebuffi et al. (2017)), and *icarl_contrary* (select the samples farthest from their prototypes). As shown in Table 4, we report the pseudo-label accuracy (Acc) and average feature variance (Var) of replayed samples, as well as the final classifier accuracy of PLRD. We can see that *icarl* has the highest pseudo-label accuracy while *icarl_contrary* has the largest sample variance and is inclined to diversity. However, PLRD under the *random* strategy has the highest OOD and ALL accuracy. This demonstrates that neither accuracy nor diversity alone leads to better performance. CGID needs to strike a balance between diversity and accuracy of the replayed samples.

**Quantity of Replayed Samples** Fig 7 illustrates the effect of replaying different numbers of previous examples. It is evident that replaying more previous examples leads to higher accuracy. Compared with replaying no examples ($n$=0), storing just one example for each old class ($n$=1) significantly improves accuracy, demonstrating that replaying old samples is crucial. In addition, PLRD outperforms the baselines significantly when $n \leq 10$, proving PLRD's effectiveness with few-shot samples replay. However, when all previous examples are replayed ($n$=ALL), PLRD performs slightly worse than E2E. We believe this is because PLRD's anti-forgetting mechanism limits learning new classes lightly, and replaying all previous examples deviates from the setting of CGID.

## 5.4 Ablation Study

As reported in Table 5, we perform ablation study to investigate the effect of each learning objective

on the PLRD framework. When removing $\mathcal{L}_{fd}$, the performance declines significantly in both IND and OOD classes. This suggests that forgetting is one of the main challenges faced by CGID, and mitigating forgetting can bring positive effects to continual OOD learning stage by retaining prior knowledge. In addition, removing $\mathcal{L}_{ins}$ and $\mathcal{L}_{pcl}$ respectively leads to a certain degree of performance decline, indicating that prototype and instance-level contrastive learning are helpful for OOD discovery and relieving OOD noise. Finally, only retaining the $\mathcal{L}_{ce}$ of PLRD will result in the largest accuracy decline, proving the importance of multiple optimization objectives in PLRD.

## 5.5 Estimate the Number of OOD intents

In the previous experiments, we assumed that the number of new OOD classes at each stage is predefined and is ground-truth. However, in practical applications, the number of new classes usually needs to be estimated automatically. We adopt the same estimation algorithm as Zhang et al. (2021); Mou et al. (2022b). [8]. Since the estimation algorithm is based on sample features, we use the model itself as the feature extractor at the beginning of each OOD learning stage. As shown in Table 6, when the estimated number of classes is inaccurate, the performance of all methods declines to some extent. However, PLRD can estimate the number most accurately and achieve the best performance. Then, in order to align different methods, we consistently use the frozen model after finishing the IND training stage as the feature extractor for subsequent stages. With the same estimation quality, PLRD still significantly outperforms each baseline, demonstrating that PLRD is robust.

---

[8]We leave the details of the algorithm in Appendix C.

# 6 Challenges

Based on the above experiments and analysis, we comprehensively summarize the unique challenges faced by CGID :

**Conflicts between different sub-tasks** In CGID, the discovery and classification of new OOD classes tend towards different features, and learning new OOD classes interfere with existing knowledge about old classes inevitably. However, preventing forgetting will lead to model rigidity and is not conducive to the learning of new classes.

**OOD noise accumulation and propagation** In the continual OOD learning stage, using pseudo-labeled OOD samples with noise to fine-tune the model as well as replaying samples with noise will cause the noise to accumulate and spread to the learning of new OOD samples in future stages. This will potentially affect the model's ability to learn effectively from new OOD samples in subsequent stages of learning.

**Fine-grained OOD classes** Section 4.3 indicate that fine-grained data leads to high forgetting and poor performance. We believe this is because fine-grained new classes and old classes are easily confused, which brings serious conflicts between new and old tasks.

**Strategy for replayed samples** The experiment in Section 5.3 shows that CGID needs to consider the trade-off between replay sample diversity and accuracy as well as the trade-off between quantity of replayed samples and user privacy.

**Continual quantity estimation of new classes** Section 5.5 shows that even minor estimation errors for each stage can accumulate over stages, leading to severely biased estimation and deteriorated performance.

# 7 Related Work

**OOD Intent Discovery** OOD Intent Discovery aims to discover new intent concepts from unlabeled OOD data. Unlike simple text clustering tasks, it considers how to leverage IND prior to enhance the discovery of unknown OOD intents. Lin et al. (2020) use OOD representations to compute similarities as weak supervision signals. Zhang et al. (2021) propose an iterative method, DeepAligned, that performs representation learning and clustering assignment iteratively while Mou et al. (2022c) perform contrastive clustering to jointly learn representations and clustering assignments. Nevertheless, it's essential to note that OOD intent discovery primarily focus on unveiling new intents, overlooking the integration of these newfound, unknown intents with the existing, well-defined intent categories.

**Generalized Intent Discovery** To overcome the limitation of OOD intent discovery that cannot expand the existing classifier, Mou et al. (2022b) proposed the Generalized Intent Discovery (GID) task. GID takes both labeled IND data and unlabeled OOD data as input and performs joint classification over IND and OOD intents. As such, GID needs to discover semantic concepts from unlabeled OOD data and learn joint classification. However, GID can only perform one-off OOD learning stage and requires full data of known classes, severely limiting its practical use. Therefore, we introduce Continual Generalized Intent Discovery (CGID) to address the challenges of dynamic and continual open-world intent classification.

**Class Incremental Learning** The primary goal of class-incremental learning (CIL) is to acquire knowledge about new classes while preserving the information related to the previously learned ones, thereby constructing a unified classifier. Earlier studies (Ke et al., 2021; Geng et al., 2021; Li et al., 2022) mainly focused on preventing catastrophic forgetting and efficient replay in CIL. However, these studies assumed labeled data streams, whereas in reality large amounts of continuously annotated data are hard to obtain and the label space is undefined. Unlike CIL, CGID charts a distinct course by continuously identifying and assimilating new classes from unlabeled OOD data streams. This task presents a set of formidable challenges compared to conventional CIL.

# 8 Conclusion

In this paper, we introduce a more challenging yet practical task as Continuous Generalized Intent Discovery (CGID), which aims at continuously and automatically discovering new intents from OOD data streams and incrementally extending the classifier, thereby enabling dynamic intent recognition in an open-world setting. To address this task, we propose a new method called Prototype-guided Learning with Replay and Distillation (PLRD). Extensive experiments and qualitative analyses validate the effectiveness of PLRD and provide insights into the key challenges of CGID.

## Limitations

This paper proposes a new task as Continual General Intent Discovery (CGID) aimed at continually and automatically discovering new intents from unlabeled out-of-domain (OOD) data and incrementally expand them to the existing classifier. Furthermore, a practical method Prototype-guided Learning with Replay and Distillation (PLRD) is proposed for CGID. However, there are still several directions to be improved: (1) Although PLRD achieves better performance than each baseline, the performance still has a large gap to improve compared with the theoretical upper bound of a model without forgetting previous knowledge. (2) In addition, all baselines and PLRD use a small number of previous samples for replay. The CGID method without utilizing any previous samples is not explored in this paper and can be a direction for future work. (3) Although PLRD does not generate additional overhead during inference, it requires maintaining prototypes and a frozen copy of the encoder during training, resulting in additional resource occupancy. This can be further optimized in future work.

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

## A  Datasets

We present detailed statistics for the original dataset Banking and CLINC in Table 7. Then, we show statistics of the CGID datasets that are constructed based on Banking and CLINC in Table 8. Since we conduct three random partitions of classes under each OOD ratio and Banking is class-imbalanced, we report the the average number of samples.

## B  Implementation Details

To ensure a fair comparison for PLRD and all baselines, we consistently use the pre-trained BERT model (BERT-base uncased [9], with 12 layer transformer) as the network backbone and add a pooling layer to obtain the intent representation (dimension = 768). In addition, we freeze all but the last transformer layer parameters to achieve better performance with BERT backbone, and speed up the training process as suggested in Lin et al. (2020). We use the same IND training phase and Memory mechanism for all methods. In the IND training stage, following Zhang et al. (2021), we adopt Adam with linear warm-up as the optimizer, with a batch size of 128 and a learning rate of 5e-5, and select the best checkpoint according to the accuracy of validation set.

During the continual OOD training stages, for the CGID baseline, we follow the hyperparameter settings of the original method as much as possible. Specifically, for DeepAligned and K-means, following Zhang et al. (2021), we adopt Adam with linear warm-up as the optimizer and set the training batch size to 128, the learning rate to 5e-4. In addition, we set the weight coefficient $\lambda$ =3 of the cross-entropy loss corresponding to the replay samples, which is obtained by grid search $\lambda \in \{1, 2, 3, 4, 5\}$. For E2E, following Mou et al. (2022b), we use SGD with momentum as the optimizer with linear warm-up and cosine annealing (warm-up ratio of 10%, momentum of 0.9, maximum learning rate of 0.1, and weight decay of 1.5e-4). In addition, the temperature coefficient of cross-entropy is 0.1

---

[9]https://github.com/google-research/bert

and multi-head clustering (number of heads is 4) is used to improve performance for E2E.

For all experiments of PLRD, we also use SGD with momentum as the optimizer (warm-up ratio is 10%, momentum is 0.9, maximum learning rate is 0.01, and weight decay of 1.5e-4). Class prototype embedding (dimension=128) is obtained by a linear projection layer through the output feature (dimension=768) of the encoder. We set the temperature $\tau$ of prototype and instance-level contrastive learning to 0.5. For the construction of the augmented examples, we set the dropout value is 0.5. Following (Mou et al., 2022b), we also calibrate the logit $l_i^{new}$ by the SK algorithm (Cuturi, 2013) when assigning prototypes. We set the corresponding hyperparameters such as number of iteration is 3 and $\epsilon$=0.05 as same as E2E. When OOD ratio=40% or 60%, the moving average coefficient $\gamma$=0.7, and when OOD ratio=80%, $\gamma$=0.9. We believe that high OOD ratio will lead to high forgetting, while larger moving average coefficient can mitigate this by helping the model remember learned prototypes.

For all methods, we train 200 epochs for each OOD learning stage to achieve sufficient convergence. The trainable model parameters of PLRD are almost consistent with the baseline (approximately 9.1M). However, PLRD adopts prototype learning and a frozen BERT for knowledge distillation, , which leads to additional memory occupation and training computation. It should be noted that PLRD only needs the classifier branch in the inference stage, so there is no additional computational and spatial overhead. All experiments use a single Nvidia RTX 3090 GPU (24 GB memory).

## C  The algorithm of estimating the number of new classes

We follow the estimation algorithm (Zhang et al., 2021) to estimate the number of new intents for each OOD stage. Specifically, We assign a big $K'$ as the number of clusters (In this paper, it is twice the number of ground-truth classes) at the beginning of each OOD stage. As a good feature initialization is helpful for partition-based methods (e.g., K-means) (Platt et al., 1999), we use the encoder $f(\cdot)$ to extract intent features of all new training examples. Then, we perform K-means with the extracted features. We suppose that real clusters tend to be dense even with $K'$, and the size of more confident clusters is larger than some threshold $t$. Therefore, we drop the low confidence

| Dataset | Classes | Training | Validation | Test | Vocabulary | Length (max / mean) |
|---------|---------|----------|------------|------|------------|---------------------|
| Banking | 77 | 9003 | 1000 | 3080 | 5028 | 79/11.91 |
| CLINC | 150 | 18000 | 2250 | 2250 | 7283 | 28/8.31 |

Table 7: Detailed statistics of original Banking and CLINC datasets.

| Dataset | OOD Ratio | New Classes | All Classes | Domains | Train samples | Val Samples | Test samples |
|---------|-----------|-------------|-------------|---------|---------------|-------------|--------------|
| | 40% | 47/10/10/10 | 47/57/67/77 | 1/1/1/1 | 5445/1136/1222/1200 | 604/127/136/133 | 1880/2280/2680/3080 |
| Banking | 60% | 32/15/15/15 | 32/47/62/77 | 1/1/1/1 | 3612/1819/1848/1724 | 401/202/206/191 | 1280/1880/2480/3080 |
| | 80% | 17/20/20/20 | 17/37/57/77 | 1/1/1/1 | 1919/2432/2329/2323 | 214/271/258/257 | 680/1480/2280/3080 |
| | 40% | 90/20/20/20 | 90/110/130/150 | 10/10/10/10 | 10800/2,400/2,400/2,400 | 1350/300/300/300 | 1350/1650/1950/2250 |
| CLINC | 60% | 60/30/30/30 | 60/90/120/150 | 10/10/10/10 | 7200/3600/3600/3600 | 900/450/450/450 | 900/1350/1800/2250 |
| | 80% | 30/40/40/40 | 30/70/110/150 | 10/10/10/10 | 3600/4800/4800/4800 | 450/600/600/600 | 450/1050/1650/2250 |

Table 8: Detailed statistics of the CGID datasets, where x/x/x/x respectively represent data under the stage 0/1/2/3.

cluster which size smaller than $t$, and calculate $K$ with:

$$K = \sum_{i=1}^{K'} \delta(|S_i| \geq t) \qquad (10)$$

where $|S_i|$ is the size of the $i^{th}$ produced cluster, and $\delta(\cdot)$ is an indicator function. It outputs 1 if the condition is satisfied, and outputs 0 if not. Notably, we assign the threshold $t$ as the expected cluster mean size $\frac{N}{K'}$ in this formula.