# OpenReview forum: "Continual Generalized Intent Discovery: Marching Towards Dynamic and Open-world Intent Recognition"
_EMNLP/2023/Conference — EMNLP 2023 Findings_

### Official Review · Reviewer_3pu5 · 2023-08-03

**Typos Grammar Style And Presentation Improvements:** There some typos in Eq.(4).
**Soundness:** 3

**Excitement:**

2: Mediocre: This paper makes marginal contributions (vs non-contemporaneous work), so I would rather not see it in the conference.

**Paper Topic And Main Contributions:**

Due to the existing limitations of the Generalized Intent Discovery (GID) task, which involves a single-stage learning process, specifically considering only one stage of out-of-domain (OOD) learning, and requires joint training with all previous data, it fails to address the continuous flow of OOD intents in an open-world scenario. To tackle this issue, this paper proposes a novel task paradigm called Continual Generalized Intent Discovery (CGID), which dynamically recognizes OOD intent data streams and extends them to the classifier without relying on previous data. The paper also introduces the PLRD (Prototype-guided Learning with Replay and Distillation) method to address this task.
The main contributions of this paper include:
1. Introduction of CGID: The paper proposes a new paradigm, CGID, to solve the problem of dynamic intent recognition in an open-world scenario. It also introduces evaluation criteria and baselines to assess the effectiveness of the proposed approach.
2. PLRD Method: To address the CGID paradigm, the paper presents an effective PLRD method that leverages class prototypes to learn new samples and balances the gap between new and old tasks through data replay and feature distillation.
3. Comprehensive Experimentation and Analysis: The paper conducts thorough experiments and in-depth analysis to validate the effectiveness of the PLRD method and discusses the challenges and prospects of the CGID task.

**Questions For The Authors:**

1. Why apply the continual learning paradigm to the GID task?
2. Why is the main experiment not compared with others, and does using stage=3 fail to demonstrate the superiority of PLRD?
3. Is the majority of the paper's focus on addressing knowledge forgetting and procedural learning brought about by the continual learning paradigm, deviating somewhat from the core problem of the GID task?

**Reasons To Accept:**

1. Introduction of a New Task Paradigm: This paper introduces a novel task paradigm called CGID, aiming to address the dynamic intent recognition problem in practical dialogue systems. It extends the current Generalized Intent Discovery (GID) task and has significant implications for advancing research in this field
2. Proposal of PLRD Method: The paper presents the PLRD (Prototype-guided Learning with Replay and Distillation) method, which combines class prototypes, data replay, and feature distillation to offer a novel and effective solution for the CGID task. The comprehensive application of this method in dynamic intent discovery holds technical value.
3. Comprehensive Experimental Design: This paper includes a well-designed set of experiments to validate the effectiveness of the PLRD method and provides in-depth analysis of the challenges faced by CGID.

**Reasons To Reject:**

1. Lack of Significant Innovation: Although this paper introduces a new GID task paradigm and the PRLD method, it mainly incorporates the concept of continual learning and a solution paradigm into GID without presenting substantial innovation.
2. Insufficient Comparative Analysis: The main experiments do not compare the results with existing works but rely on self-constructed baselines, which lacks persuasiveness. Moreover, the use of a small stage=3 may not adequately validate the effectiveness of the PLRD method.
3. Limited Capability of CGID Paradigm: While the paper discusses the primary challenges faced by the CGID task, these challenges appear to be a result of incorporating continual learning into GID. It seems that the CGID paradigm itself may not be well-suited to handle the dynamic OOD intent streams in real-world scenarios due to its inherent limitations.

**Reproducibility:**

4: Could mostly reproduce the results, but there may be some variation because of sample variance or minor variations in their interpretation of the protocol or method.

**Reviewer Confidence:**

3: Pretty sure, but there's a chance I missed something. Although I have a good feel for this area in general, I did not carefully check the paper's details, e.g., the math, experimental design, or novelty.

---

> ### Author Rebuttal · Authors · 2023-08-29
>
> We appreciate your thorough review and valuable feedback. We are encouraged by your recognition of  the new CGID task paradigm, a valuable method, and comprehensive experiments. We sincerely apologize for any unclear presentation and hope this response can resolve your concerns.
>
> Q1: A new GID task with the concept of continual learning and lacks substantial novelty.
>
> A1: We aim to expound upon the innovative aspects and contributions of our paper in response to your inquiry regarding the concept of "lacks substantial novelty." Firstly, we introduce the Continual Generalized Intent Discovery (CGID) task, distinct from GID and traditional continual learning. CGID focuses on discovering new intents and enhancing classifiers within evolving unlabeled OOD data streams. Subsequently, we propose the PLRD method to tackle challenges within the CGID task, encompassing new intent discovery, forgetting, and noise propagation. This method is intricately designed with submodules such as prototype guidance, feature distillation, and data replay, culminating in an effective resolution. Lastly, through dataset creation and baseline establishment, we conduct comparative experiments that highlight significant performance improvements achieved by PLRD. This substantiates PLRD's innovative value in practical applications. In summation, while our paper builds upon existing GID and continual learning concepts to some extent, our innovation notably lies in the amalgamation of these concepts with the real-world challenges inherent to the CGID task. The proposed solution has exhibited significant improvement, further underscoring the advancement brought forth by our work.
>
> Q2: Lacks result comparison, relies on self-constructed baselines, limited stage validation.
>
> A2: In response to the points you raised, we'd like to clarify some misunderstandings to better explain our experimentation and validation process. As stated in lines 300 to 305 of the paper, this is the first work in the field of CGID, lacking direct comparisons with existing work. To address this, we thoroughly examined current GID SOTA methods and continual learning approaches, integrating them into three baseline models. Regarding your concern about the effectiveness validation at "stage=3," we have exhaustively presented the performance metrics of both PLRD and the baselines across stages 0 to 6 in Section 5.2 and Figure 6. These data conspicuously demonstrate that irrespective of the specific stage, the PLRD approach consistently outperforms all baselines. Thus, we remain steadfast in our conviction that our method is poised to manifest its efficacy and potential across the spectrum of learning stages.
>
> Q3: The limitations of the CGID paradigm may make it unsuitable for dynamic OOD intentions in real-world scenarios.
>
> A3: The core design objective of the CGID is to facilitate the ongoing discovery and learning of new intents within a dynamic and open environment, thereby enhancing its adaptability to the continually evolving user intents in the real world. Although the integration of continual learning into the GID framework may introduce certain constraining factors, it is our assertion that the presence of such limiting factors does not constitute an inherent failure of the CGID Paradigm itself; rather, it presents a significant challenge necessitating in-depth investigation and resolution. We firmly believe this to be a domain of continuous evolution. Future research endeavors can delve further into the exploration of strategies to more effectively address these issues to adeptly tackle the challenges of dynamic intent recognition within an open-world context.
>
> Q4: Why apply the continual learning paradigm to the GID task?
>
> A4: In summary, due to the open-ended nature of user queries, intent recognition systems need to continuously discover and learn new intents over time. As discussed in lines 52 to 74 of the paper, although the GID task has made progress in discovering and learning new intents from unlabeled data, it faces two main limitations. Firstly, GID only handles OOD queries and extends the classifier within a single phase. However, in reality, OOD data accumulates over time and requires continual learning for adaptation. Additionally, to preserve recognition of known intents while learning new ones, GID requires using all known intent data for joint training. Nevertheless, storing all previous data raises privacy concerns, as OOD samples are collected from real user queries. To address these limitations and challenges, we incorporate the concept of continual learning into GID, introducing the CGID task.
>
> Q5: The main experiment lacks comparison, stage=3 fails to show PLRD superiority.
>
> A5: Regarding your concern about the limited comparisons in our experiment, we aim to clarify some misconceptions. Firstly, as demonstrated in Table 2 and 3, we extensively compare PLRD with three strong baselines across diverse dataset partitions. Secondly, while we focused on stage=3 in the main experiment due to space constraints, we present the performance metrics of PLRD and all baselines across stages 0 to 6 in Section 5.6 and Figure 6. The results indicate that PLRD outperforms the baselines across all stages, thereby emphasizing its global effectiveness.
>
> Q6: Does the paper primarily address knowledge forgetting and dev from the GID task?
>
> A6: Although our paper introduces the CGID task addressing knowledge forgetting and procedural learning, our main focus remains on addressing the core issue of GID within continual and dynamic OOD data contexts. Firstly, given the dynamic nature of the task, addressing knowledge forgetting and procedural learning within CGID is a necessary step. Although these aspects may appear to deviate from the core problem of GID, they emerge as inherent challenges when attempting to extend the GID framework to handle real-world scenarios. By addressing these challenges, we enhance the applicability of the GID framework, rendering it more effective in managing evolving user intents. Secondly, distinct from traditional continual learning, our work centers on solving continuous learning with unlabeled OOD data. This brings about unique challenges, such as balancing diversity in OOD replay samples and pseudo-label accuracy, as well as the accumulation and propagation of OOD noise. As a result, specific approaches are needed to address these challenges. We hope these explanations can provide clearer insight into the focal points of our work.
>
> Q7: Typos Grammar Style And Presentation Improvements.
>
> A7: We will enhance our grammar and expression in the revised version based on your feedback.

---

### Official Review · Reviewer_Ldu4 · 2023-08-04

**Typos Grammar Style And Presentation Improvements:** NA
**Soundness:** 3

**Excitement:**

3: Ambivalent: It has merits (e.g., it reports state-of-the-art results, the idea is nice), but there are key weaknesses (e.g., it describes incremental work), and it can significantly benefit from another round of revision. However, I won't object to accepting it if my co-reviewers champion it.

**Missing References:**

NA

**Paper Topic And Main Contributions:**

This work first formulates the problem of Continual Generalized Intent Discovery (CGID), in contrast to traditional Generalized Intent Discovery(GID). The authors argue that Out of Domain Intent Discovery(OOD) in the real world is a continuous process, where new OODs continuously emerge, but traditional GID systems only do intent discovery in one phase only.

Traditional GIDs also need to rely on retraining the system with all of In Domain(ID) data along with the new OOD data, to mitigate the problem of forgetting previous learning. This shortcoming of GIDs has privacy concerns, as the data used by these systems is highly personal, and storing it for later recall can have privacy breaches.

The authors provide a system to do Continual Generalized Intent Discovery. They use three modules in the system 1) Memory module: stores a subset of previous data for recall. 2)Class prototype module is responsible for generating pseudo-labels for new OOD samples 3) Feature distillation is responsible for balancing between new and old data for alleviating catastrophic forgetting of old classes.

**Questions For The Authors:**

What if, instead of using previous known dataset(which has privacy concerns), system passes randomly generated data to get the features for  knowledge distillation. i.e: to capture the previous learning, before adding new training data, we pass random inputs and capture the projections, and then use this pseudo dataset to maintain previous distribution.

**Reasons To Accept:**

-Novel Problem definition.

-Comprehensive Experimentation.

**Reasons To Reject:**

-Limited Novelty.

-Essentially the same system as traditional GIDs with limiting the size of the recall dataset.

-Still has the same privacy concerns as it also stores a subset of previous data.

**Reproducibility:**

4: Could mostly reproduce the results, but there may be some variation because of sample variance or minor variations in their interpretation of the protocol or method.

**Reviewer Confidence:**

4: Quite sure. I tried to check the important points carefully. It's unlikely, though conceivable, that I missed something that should affect my ratings.

---

> ### Author Rebuttal · Authors · 2023-08-29
>
> We appreciate your thorough review and valuable feedback. We are encouraged by your recognition of  the novel problem definition and comprehensive experimentation. We sincerely apologize for any unclear presentation and hope this response can resolve your concerns.
>
> Q1: Limited Novelty.
>
> A1: We aim to expound upon the innovative aspects and contributions of our paper in response to your inquiry regarding the concept of "Limited Novelty." Firstly, we introduce the Continual Generalized Intent Discovery (CGID) task, distinct from GID and traditional continual learning. CGID focuses on discovering new intents and enhancing classifiers within evolving unlabeled OOD data streams. Subsequently, we propose the PLRD method to tackle challenges within the CGID task, encompassing new intent discovery, forgetting, and noise propagation. This method is intricately designed with submodules such as prototype guidance, feature distillation, and data replay, culminating in an effective resolution. Lastly, through dataset creation and baseline establishment, we conduct comparative experiments that highlight significant performance improvements achieved by PLRD. This substantiates PLRD's innovative value in practical applications. In summation, while our paper builds upon existing GID and continual learning concepts to some extent, our innovation notably lies in the amalgamation of these concepts with the real-world challenges inherent to the CGID task. The proposed solution has exhibited significant improvement, further underscoring the advancement brought forth by our work.
>
> Q2: Essentially the same system as GID, with limiting data recall.
>
> A2: Regarding the matter you have raised, we seek to elucidate certain misconceptions. While CGID has been devised as an extension upon the foundation of GID, it presents distinct challenges in various facets, such as continual learning from unlabeled OOD data, catastrophic forgetting, and the accumulation and propagation of OOD noise (refer to Section 6). These challenges drove us to propose the PLRD method as a solution. The perspective you mentioned as "Essentially the same system as traditional GIDs with limiting recall" actually corresponds to the baselines in this paper (refer to Section 4.2). Our proposed PLRD, incorporating key components including prototype-guided learning and feature distillation, effectively addresses these challenges, allowing for the discovery of new categories while significantly mitigating forgetting. Thorough experimentation confirms the substantial improvements of PLRD over the baselines (refer to Section 4.3, 5.2, 5.3).
>
> Q3: Still has privacy concerns as it also stores a subset of previous data.
>
> A3: As illustrated in Section 3.2, we introduce a memory mechanism that enables PLRD to balance the learning of new and old tasks by replaying previous samples. This replay process necessitates the retention of an exceedingly small subset of samples (in our experimentation, only 5 samples are stored per class). Consequently, in contrast to GID methods which require the storage of all prior data, PLRD significantly reduces the volume of stored data, thereby exhibiting notable advantages in terms of privacy. However, it is imperative to candidly acknowledge that privacy concerns are not entirely resolved despite our efforts. As indicated in Section 5.3 and Figure 7, both PLRD and all baselines encounter a sharp decline in performance when prior data is completely unused. We delve into a thorough consideration of the trade-off between privacy concerns and performance aspects, striving to identify an equilibrium point. The introduction of the PLRD method serves not only to provide a privacy-conscious mode of learning but also takes into account the long-term stability of task performance. For methods that completely eliminate the need for prior data storage, we hope to keep it for further exploration.
>
> Q4: Privacy-safe approach: replace known data with random inputs for knowledge distillation.
>
> A4: We greatly appreciate your attention and suggestions regarding our paper. We regard the utilization of randomly generated data for feature acquisition followed by knowledge distillation as an intriguing approach, potentially offering a comprehensive solution to privacy concerns arising from the retention of prior data.  However, this method may encounter the following challenges: Firstly, relying on randomly generated data might not accurately capture the distribution of real-world data, leading to disparities between synthetic and actual data that could affect the success of knowledge distillation. Secondly, using synthetic data for knowledge distillation could result in model overfitting and introduce unknown biases. Therefore, it's crucial to ensure a reasonable similarity between the generated data and actual data in the feature space. For example, focusing on feature points near class prototypes might yield data more representative of actual instances. Despite these obstacles, we believe this idea deserves further investigation.

---

### Official Review · Reviewer_fQPy · 2023-08-08

**Soundness:** 3

**Excitement:**

3: Ambivalent: It has merits (e.g., it reports state-of-the-art results, the idea is nice), but there are key weaknesses (e.g., it describes incremental work), and it can significantly benefit from another round of revision. However, I won't object to accepting it if my co-reviewers champion it.

**Paper Topic And Main Contributions:**

This paper focuses on the task of Out-of-Domain Discovery for Intent classification systems. Generalized Intent Discovery systems aim to discover new intents from OOD queries and integrate them into the system over time. The authors argue that existing systems are atomic in nature i.e the classifier is expanded in a single stage as opposed to being continual and expanding naturally over time. Additionally, existing generalized intent discovery systems require access to data from all prior stages in order to maintain coverage of known intents when expanding. As such the authors introduce Continual Generalized Intent Discovery (CGID) task which aims to continually and automatically discover OOD intents from OOD data and expand it into the in-domain taxonomy. They propose a method called Prototype guided Learning with Replay and Distillation (PLRD) for solving this task and construct a series of baselines and datasets for evaluating the CGID task.

**Questions For The Authors:**

- How does the memory module alleviate the privacy concern issues if it stores prior training samples for both IND and OOD stages?

**Reasons To Accept:**

 - PLRD consistently outperforms baselines
- Detailed analysis examining performance across multiple stages and datasets
- Construction of new datasets for evaluating dynamic OOD detection.

**Reasons To Reject:**

- Arguments are not sufficiently justified or convincing. After reading the introduction several times it was unclear as to why the task of Continual Generalized Intent Discovery is needed, making this work seem very incremental in nature. The authors claim that accessing data from the prior stage raises privacy concerns but do not expound or justify this. Also, the argument for the timeliness of classifier expansion mentioned isn't actually explored in the method proposed.

**Reproducibility:**

4: Could mostly reproduce the results, but there may be some variation because of sample variance or minor variations in their interpretation of the protocol or method.

**Reviewer Confidence:**

4: Quite sure. I tried to check the important points carefully. It's unlikely, though conceivable, that I missed something that should affect my ratings.

**Typos Grammar Style And Presentation Improvements:**

- Line 56 - 57: "GID only considers single stage of OOD discovery and classifier expansion.": Grammar is incorrect. Also, this is very unclear when reading it. Line 56 - 65 needs rewriting to be more crisp and concise. Given that this is the main argument in your paper, having the reader guess what this means is less than ideal.
- The mixture of color schemes and shapes across diagrams is confusing. Choose a consistent color scheme and legend, the inconsistency makes it harder the follow.

---

> ### Author Rebuttal · Authors · 2023-08-29
>
> We appreciate your thorough review and valuable feedback. We are encouraged by your recognition of  PLRD's superior performance, comprehensive multi-stage analysis, and new dataset creation. We sincerely apologize for any unclear presentation and hope this response can resolve your concerns.
>
> Q1: Insufficiently justified arguments and unclear necessity of task.
>
> A1: We apologize for any inconvenience caused by unclear expressions.  Due to the open-ended nature of user queries, intent recognition systems need to continually adapt and learn new intents over time. As outlined in lines 52 to 74 of the paper, while progress has been made in discovering and learning new intents, the existing GID task has limitations in handling OOD data in a single step and requiring all past data. To address this, we introduce the CGID task to overcome these constraints. As discussed in lines 67-69, since OOD samples arise from authentic user queries that may contain sensitive information, storing and accessing prior-stage data could raise user privacy concerns. Our proposed PLRD method allows the model to maintain the classification capability of known intents with minimal storage of past data, significantly mitigating privacy issues. Regarding the timeliness of classifier expansion, as depicted in Figure 1, traditional GID methods severely constrain the flexibility of classifier expansion due to their single-stage learning approach. In contrast, PLRD can incrementally learn new knowledge from unlabeled OOD data in a multi-stage manner, thereby requiring only a small accumulation of OOD data to learn new categories and expand the classifier. This facilitates timely expansion.
>
> Q2: How does the memory module alleviate the privacy concern issues if it stores prior training samples?
>
> A2: As illustrated in Section 3.2, we introduce a memory mechanism that enables PLRD to balance the learning of new and old tasks by replaying previous samples. This replay process necessitates the retention of an exceedingly small subset of samples (in our experimentation, only 5 samples are stored per class). Consequently, in contrast to GID methods that require the storage of all prior data, PLRD significantly reduces the volume of stored data, thereby exhibiting notable advantages in terms of privacy. However, it is imperative to candidly acknowledge that privacy concerns are not entirely resolved despite our diligent efforts. As indicated in Section 5.3 and Figure 7, both PLRD and the baselines encounter a sharp decline in performance when prior data is completely unused. We delve into a thorough consideration of the trade-off between privacy concerns and performance aspects, striving to identify an equilibrium point. The introduction of the PLRD method serves not only to provide a privacy-conscious mode of learning but also takes into account the long-term stability of task performance. As for methods that completely eliminate the need for prior data storage, we intend to leave them for further exploration.
>
> Q3: Typos Grammar Style And Presentation Improvements.
>
> A3: We will enhance our grammar and expression in the revised version based on your feedback.

---

### Meta-Review · Area_Chair_8mcG · 2023-09-24

**Recommendation:** 3

**Metareview:**

This paper introduces the task of Continual Generalized Intent Discovery (CGID), tackling existing limitations of Generalized Intent Discovery (GID).  Specifically, the paper proposes the Prototype-guided Learning with Replay and Distillation (PLRD) to address this task.  The reviewers appreciate the introduction of a new task that adds continual learning to the GID task, the proposed PLRD model for solving this task, and the experiments that demonstrate the model's effectiveness.  However, the work is considered to be somewhat lacking in novelty.

---

### Meta-Review · Senior_Area_Chairs · 2023-10-05

**Recommendation:** 3

**Metareview:**

meta review

---

### Decision · Program_Chairs · 2023-10-07

**Decision:**

Accept-Findings

**Comment:**

This paper introduces the task of Continual Generalized Intent Discovery (CGID), tackling existing limitations of Generalized Intent Discovery (GID).  Specifically, the paper proposes the Prototype-guided Learning with Replay and Distillation (PLRD) to address this task.  The reviewers appreciate the introduction of a new task that adds continual learning to the GID task, the proposed PLRD model for solving this task, and the experiments that demonstrate the model's effectiveness.  However, the work is considered to be somewhat lacking in novelty.|meta review